# Stereoisomers of Colourless Carotenoids from the Marine Microalga *Dunaliella salina*

**DOI:** 10.3390/molecules25081880

**Published:** 2020-04-18

**Authors:** Laura Mazzucchi, Yanan Xu, Patricia Harvey

**Affiliations:** School of Science, Faculty of Engineering and Science, University of Greenwich, Central Avenue, Chatham Maritime, Kent ME4 4TB, UK; L.Mazzucchi@greenwich.ac.uk (L.M.); Y.Xu@greenwich.ac.uk (Y.X.)

**Keywords:** phytoene, phytofluene, isomers, carotenoids, *Dunaliella salina*, chlorpropham, norflurazon, red light

## Abstract

Carotenoids comprise a diverse range of naturally occurring stereoisomers, which differ in their physico-chemical properties. Their biosynthesis begins with phytoene, which is a rarity among carotenoids because it is colourless. Phytoene is sought after as a skin protectant against harmful UV range B (290–320 nm) and C (100–290 nm) light, and as a natural skin-whitening agent and is synthesized from geranylgeranyl diphosphate. Geranylgeranyl diphosphate is catalysed by phytoene synthase and phytoene desaturase to phytoene and phytofluene, respectively. The subsequent steps involve desaturation, isomerisation and cyclisation reactions to form α- and β-carotene stereoisomers, via *all-trans* lycopene. The marine microalga *Dunaliella salina* is the richest source of β-carotene, but it can accumulate phytoene and phytofluene as well. In the present study, different analytical tools including High-Performance Liquid Chromatography (HPLC), Ultra-Performance Convergence Chromatography (UPC^2^-MS) and Nuclear Magnetic Resonance (NMR) were used to characterize and quantify the phytoene isomeric configurations in *D. salina* in order to explore both the feasibility of *D. salina* as a cell factory for phytoene production and to gain new insight into the carotenoid synthesis pathway in *D. salina. D. salina*, similar to tomato, produced predominantly *15-cis* phytoene isomer (>98%) and a trace amount of *all-trans* phytoene (<2%). High light stress, red light stress, or use of a phytoene desaturase inhibitor or a mitotic disrupter herbicide led to the accumulation of *15-cis* phytoene but not *all-trans* phytoene. *9-cis* phytoene was not detected in any of the extracts of *D. salina* biomass. Our main findings suggest that *15-cis* phytoene is the most abundant isomer in *D. salina* and that it is subject to a series of isomerisation and desaturation reactions to form *all-trans* and *9-cis* β-carotene.

## 1. Introduction

Carotenoids are lipophilic compounds with a common C40 backbone of isoprenoid units and are naturally synthesized by photosynthetic organisms and some non-photosynthetic bacteria and fungi [1,2]. The high number of conjugated double bonds in carotenoid molecules contributes to their strong antioxidant capacity, and therefore they can potentially protect humans against ageing and diseases that are caused by harmful free radicals. The colourless carotenoids, phytoene ((6*E*,10*E*,14*E*,16*E*,18*E*,22*E*,26*E*)-2,6,10,14,19,23,27,31-octamethyldotriaconta-2,6,10,14,16,18,22,26,30-nonaene) and phytofluene ((6*E*,10*E*,12*E*,14*E*,16*E*,18*E*,22*E*,26*E*)-2,6,10,14,19,23,27,31-octamethyldotriaconta-2,6,10,12,14,16,18,22,26,30-decaene) are of particular interest: phytoene, the progenitor in the carotenoid synthesis pathway, is able to absorb both hazardous UV range B (290–320 nm) and C light (100–290 nm) and phytofluene, to absorb UV range A (315–400 nm). They may protect the skin against erythema, premature skin aging and skin cancer [3]. Phytoene has also been reported to be anti-inflammatory, hepato-protective and to prevent several other types of cancers [3,4]. Ultimately, their colourless features make phytoene and phytofluene valuable as food additives and for cosmetic products [4,5]. Phytoene and phytofluene are found in higher plants such as bell pepper, apricot, melon, orange and tomato [4]. They have also been detected in numerous cyanobacteria and microalgae. However, apart from tomatoes, the amount found in most of the organisms is small [4,6].

It is generally considered that carotenoid biosynthesis in photosynthetic plants and algae takes place in the chloroplast with some specific steps in the cytoplasm [7]. The pathway involves a series of desaturation and isomerization steps to form a diverse range of stereoisomers (Figure 1). The first committed step in carotenoid biosynthesis is the chain-elongating condensation reaction between two molecules of geranylgeranyl diphosphate (C20) to form *15-cis* phytoene (C40). This step is catalysed by the enzyme phytoene synthase (PSY) and is followed by a two-step desaturation reaction catalysed by *15-cis*-phytoene desaturase (PDS) with *9,15-di-cis*-phytofluene (equation 1), and 9,9’,15-*tri*-*cis*-ζ-carotene (equation 2) as end products (R = C_53_H_80_O_2_ plastoquinone), and plastoquinone as hydrogen acceptor [8,9,10,11].

C_40_H_64_ + 2 R 
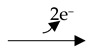
 C_40_H_62_ + 2 RH_2_(1)

C_40_H_62_ + 2 R 
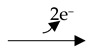
 C_40_H_60_ + 2 RH_2_(2)

The subsequent steps involve further desaturation and isomerization reactions catalysed by ζ-carotene desaturase (ZDS) and carotenoid isomerase (CrtISO) respectively, to form *all-trans* lycopene [8,9,12]. *All-trans* lycopene is cyclized into either β-carotene or α-carotene catalysed by different groups of lycopene cyclases (LYC). Xanthophyll products, which include zeaxanthin, antheraxanthin, violaxanthin, lutein and their derivatives, are formed via the hydroxylation of β-carotene or α-carotene [7].

The halotolerant green microalga *Dunaliella salina* is one of the richest sources of carotenoids, especially *all-trans-* and *9-cis* β-carotene. However, it will also accumulate phytoene and phytofluene in the presence of a phytoene desaturase (PDS) inhibitor [13,14]. *D. salina* is easy to culture, as it requires light and a few other nutrients (nitrogen, phosphate and salts) to grow. For these reasons, it is considered a promising species to produce carotenoids [15].

Bleaching herbicides such as norflurazon are known to boost phytoene accumulation in the cells of *D. salina* and other higher plants, by inhibiting PDS, which prevents the conversion of phytoene to phytofluene [14,16,17]. However, little information has been provided on the isomer composition of the accumulated phytoene and available data to date are contradictory. Ebenezer [18] for example showed by using Nuclear Magnetic Resonance (NMR) methods that the main phytoene isomer, which accumulated in norflurazon-treated *D. bardawil* cultures *(**Dunaliella bardawil* is a variant of *Dunaliella salina**)*, was the *15-cis* isomer, whilst Werman et al. [6] and Ben-Amotz et al. [13] using High Performance Liquid Chromatography (HPLC) techniques, reported that the phytoene-rich *D. bardawil* powder used in their study contained the *9-cis* phytoene isomer together with *all-trans* phytoene. This leads to an unresolved point in regard to the carotenoid biosynthetic pathway of *D. salina*: whether the high quantity of *9-cis* β-carotene derives directly from *9-cis* phytoene, as proposed in [13], or from the isomerization of *all-trans* β-carotene, as proposed by Davidi et al. [19] and by Xu and Harvey [14,20]. Resolving the identity of the phytoene isomers produced by *D. salina* could provide new insight into the carotenoid synthesis pathway and clarify this point.

The ratio of absorption peak heights at defined wavelengths, which defines fine structure, is useful for distinguishing carotenoids and their isomers and HPLC coupled with UV-Vis detection appears to be the most common method adopted for their study [21]. However, most studies have examined phytoene extracted from tomato. In ripening tomato, *all-trans* phytoene usually has a higher fine structure than *15-cis* phytoene [22]. NMR analysis is necessary to discriminate at a structural level between different isomers. The values of specific chemical shifts for the *cis-* and *trans-* isomeric forms of phytoene have been assigned in previous studies [18,23,24].

The main objective of this work was to resolve the phytoene isomers in *D. salina* and to gain new insight into the carotenoid synthesis pathway of *D. salina*. Methods of sample preparation from microalgal biomass and NMR methods were developed to characterize the phytoene isomers. The effects of cultivation conditions, light wavelength and light intensity, and herbicide treatment with two different classes of herbicides, chlorpropham and norflurazon, on carotenoid synthesis and phytoene isomer accumulation were also investigated.

## 2. Results

### 2.1. Identification of Phytoene Isomers in D. salina with HPLC

In the present study, HPLC analysis was carried out to identify the phytoene isomers in *D. salina* cultures. Method parameters were tuned to reduce the shifts in retention time of the phytoene peaks among runs. The phytoene standard purchased from Sigma is composed of a mixture of 6 isomers (Figure 2). Peak 3 of the standard matched with the phytoene isomer from tomato extract, which is widely considered as *15-cis* phytoene. Peak 3 is also the closest peak to that from *D. salina* extracts, suggesting that phytoene accumulated in *D. salina* is most likely to be *15-cis* phytoene as well.

A flow rate of 0.5 mL/min under isocratic solvent conditions (MeOH/MTBE- 88/10) was used to elute stereo-mutated tomato extract and phytoene standard to reveal the details of the elution peaks (Figure 3). The tomato extract stereo-mutated showed the presence of a second peak eluting after the *15-cis* isomer, which was identified as *all-trans* phytoene, in accordance with previous work by Melendez-Martinez et al. [22]. This peak matched with peak 5 of the phytoene standard and peak 2 of *D. salina* extract shown previously in Figure 2, suggesting they are *all-trans* phytoene isomers.

Although both *15-cis* and *all-trans* phytoene isomers gave absorption maxima at 275, 285 and 297 nm, they had slightly different absorption spectra (Figure 4). *15-cis* phytoene showed a loss in fine structure in comparison with the *all-trans* phytoene in all the extracts, which agrees with the work by Melendez-Martinez et al. [22] and Britton et al. [25].

### 2.2. Phytoene Accumulation in D. salina Treated by Chlorpropham or Norflurazon

Use of either chlorpropham or norflurazon leads to over production of phytoene in *D. salina* [13,14]. Here, we treated the cultures with 20 µM chlorpropham or 5 µM norflurazon and found that the percentage abundance of the *15-cis* phytoene in total phytoene was 99.80 ±0.08, 99.80 ± 0.07 and 99.78 ± 0.07 for the control cultures, chlorpropham-treated cultures and norflurazon-treated cultures respectively. *15-cis* phytoene was therefore the predominant isomer that accumulated with herbicide treatment (Figure 5).

### 2.3. Analysis of Phytoene with UPC^2^-MS

The presence of the *15-cis* phytoene in the norflurazon- and chlorpropham-treated *D. salina* extracts was evaluated with UPC^2^ analysis coupled with mass spectrometry, using *15-cis* phytoene extracted from tomato as a standard. Figure 6 shows the UPC^2^ chromatograms of phytoene extracted from tomato and *D. salina* cultures treated with either chlorpropham or norflurazon. The chromatograms could be superimposed on one another (RT chlorpropham extract = 2.50 min; RT norflurazon extract = 2.49 min; RT tomato = 2.49 min). Mass accuracy analysis confirmed the presence of phytoene in all the extracts (Figure 7). Protonated ions [M + H]^+^ corresponding to phytoene were all within 10 ppm.

### 2.4. NMR Analysis of Phytoene Extracts from D. salina

In the present study, NMR analysis was carried out to prove that the main peak from *D. salina* detected with HPLC was *15-cis* phytoene. Phytoene in a carotenoid-lipophilic mixture extracted from tomato was analysed with NMR, and chemical shifts values were assigned and compared to those identified in the mixture extracted from *D. salina* biomass. Figure 8a shows the results of 2D-HSCQ obtained from the tomato extract. Proton Chemical shifts of the conjugated central part C_14_, C_15_, C_15′_, C_14′_ were as follow: δ _h14,14′_ = 6.31 ppm, δ _h15, 15′_ = 6.100. Carbon chemical shifts were H-C (14, 14′) 120.00, H-C (15, 15′) 123.18. Chemical shifts values assigned were in accordance with the literature [18,23,24]. 2D-HSCQ NMR spectra of the alga biomass are shown in Figure 8b,c. The spectra of the tomato and the alga mixtures are superimposable. Proton chemical shifts were δ _h14,14′_ = 6.31 ppm and δ _h15, 15′_ = 6.10, and δ _h14,14′_ = 6.31 ppm and δ _h15, 15′_ = 6.10 ppm for the chlorpropham and norflurazon treated alga respectively. Carbon chemical shifts were H-C (14, 14′) 120.42, H-C (15, 15′) 123.60 for both the treated algae. These results show that in the presence of either chlorpropham or norflurazon, the alga *Dunaliella salina* accumulates *15-cis* phytoene as the main isomer.

### 2.5. Colourless Phytofluene Isomers Produced by D. salina

Phytofluene isomeric forms were detected at the wavelength of 350 nm in HPLC-diode array analysis (Figure 9). Two phytofluene peaks were detected in tomato extracts. Peak 1 from tomato extracts matched with peak 1 of *D. salina* extracts (RT: (a) 16.177; (b) 16.171; (c) 16.212 and (d) 16.186) and corresponded to the major phytofluene in both tomato and *D. salina*. Phytofluene was over-accumulated in chlorpropham-treated *D. salina* cultures but not in norflurazon-treated *D. salina* cultures (Figure 9a). The second phytofluene isomeric form in tomato extracts appeared in trace amount and could only be detected in the extracts of chlorpropham treated *D. salina* cultures (RT: (a) 23.714 and RT (b) 23.674), but not in the norflurazon treated cultures or the control cultures.

### 2.6. Phytoene and Phytofluene Production in D. salina under Different Light Conditions

Previously, we have shown that light wavelengths affect the synthesis of carotenoids in *D. salina.* We found that the red light increased the cellular content of all major carotenoids and total carotenoids [14]. Figure 10 shows the concentration of phytoene isomers in *D. salina* cultures grown under red or blue light at different light intensities (200, 500 or 1000 µmol·m^−2^·s^−1^) for 48 h. The major phytoene isomer *15-cis* phytoene in *D. salina* cultures changed with light wavelengths and light intensities, while no significant difference was found in the concentration of *all-trans* phytoene under all light conditions.

Concentrations of *15-cis* phytoene in the cultures increased with light intensity under both red and blue light, and generally showed higher concentrations under red than blue light. The ratio of *15-cis*/*all-trans* phytoene increased with light intensity from 2.2 ± 0.1 to 5.9 ± 0.1 under red light and increased from 1.4 ± 0.2 to 3.9 ± 0.4 under blue light. A Pearson correlation analysis was carried out between the two phytoene isomers, and no significant correlation was found (*p* = 0.34).

Phytofluene was measured at 355 nm using HPLC. Only one phytofluene isomer was detected in the carotenoid extracts from *D. salina* biomass, and its RT corresponded to that for phytofluene from tomato extracts. The concentration of phytofluene increased with light intensity under both red and blue light, with concentrations under red light more than double the concentrations under blue light (Figure 11).

## 3. Discussion

Phytoene demand has increased recently due to its potential health benefits. In higher plants and fruits, such as tomato, phytoene is found predominantly in the *15-cis* form. However, little work has been done to characterize the phytoene isomers in microalgae, such as *D. salina,* which is a rich source of natural carotenoids, especially β-carotene with a high ratio of the valuable *9-cis* β-carotene. This work characterized phytoene isomers in *D. salina* and compared the production of phytoene in *D. salina* cultures treated with the two different classes of herbicides norflurazon and chlorpropham. After both treatments, the concentration of *15-cis* phytoene increased. Moreover, the colourless phytofluene isomers over-accumulated in chlorpropham, but not norflurazon-treated *D. salina* cultures.

Results of HPLC analysis, UPC^2^ analysis and NMR analysis each showed that *15-cis* phytoene was the main phytoene isomer that accumulated in *D. salina* (>98% of total phytoene). Furthermore, the *15-cis* phytoene increased with the treatment of both the herbicides norflurazon and chlorpropham and with increasing light intensity. Identification of *15-cis* phytoene with only trace amounts of *all-trans* phytoene is in concord with findings for ripe tomato and higher plants [8,22] and with results obtained by Ebenezer et al. for *D. bardawil* [18], but not with the results obtained by Ben-Amotz et al. [13] and Werman et al. [6], who showed that both *9-cis* phytoene and *all-trans* phytoene accumulated in *D. bardawil* treated with herbicides.

*15-cis* phytofluene has been considered by many to be the predominant phytofluene isomer (for a review see [4]). However, Koschmieder et al. [26] compared synthetic *15-cis* phytofluene and *all-trans* phytofluene with phytofluene isomers from tomato and norflurazon-treated *Dunaliella bardawil*, and on the basis of RT obtained after HPLC analysis as well as absorption maxima at 331nm, 348 nm and 366 nm, concluded that phytofluene in tomato extract was predominately *9,15-di-cis* phytofluene, which also exists in *Dunaliella bardawil* in a small proportion. They also determined that the main phytofluene isomer in norflurazon treated *D. bardawil* was *9-cis* phytofluene but did not present the isomeric configurations in untreated *D. bardawil* cultures. In contrast, our results showed that the main phytofluene isomer in both untreated and norflurazon in chlorpropham treated *D. salina* was the same as in tomato extracts, which is *9,15-di-cis* phytofluene. The *9-cis* phytofluene shown by Koschmieder et al. [26] is likely not a phytofluene peak because norflurazon inhibits PDS function and blocks the desaturation of phytoene to subsequent phytofluene products.

The fact that only *15-cis* phytoene accumulated in *D. salina* when treated with phytoene desaturase inhibitor, while no accumulation of *all-trans* phytoene was found, is consistent with *15-cis* phytoene as the most likely precursor of phytofluene. *All-trans* phytoene present in *D. salina* is likely to be formed by the isomerisation of the predominant *15-cis* phytoene and this reaction step along with the conversion of *15-cis* phytoene to phytofluene might be fast enough to occur prior to the interference of the chlorpropham herbicide action. As a second hypothesis, *all-trans* phytoene may also be formed directly from the precursor of geranylgeranyl pyrophosphate as suggested by Gregonis and Rilling [27]. An early study on a green microalga *Scenedesmus obliquus* also supports the idea that *15-cis* phytoene is converted via *15-cis* phytofluene and *15-cis* ζ-carotene into *all-trans*-gz-carotene and *trans*-bicyclic carotenoids such as α-carotene and β-carotene [28]. Our results show that *15-cis* phytoene is the main stereoisomer produced by *D. salina* and that it is over-accumulated with red light and with the use of herbicides. Our results also support the recent finding that *9-cis* β-carotene is formed by isomerisation of *all-trans* β-carotene with the function of a *9-cis*/*all-trans* β-carotene isomerase [19,20], rather than synthesised from the precursor of *9-cis* phytoene.

## 4. Materials and Methods

### 4.1. Algal Cultivation

*D. salina* strain CCAP 19/41 (PLY DF15) was obtained from the Marine Biological Association, Plymouth, UK (MBA). Algae were cultured as described in previous work [14]. Inoculum was prepared by adding 10 mL of the culture stock of *D. salina* to 250 mL of fresh media until the culture reached the exponential phase. Algae were cultured in 500 mL Modified Johnsons Medium containing 1.5 M NaCl and 10 mM NaHCO_3_ in an illuminated incubator (Varicon Aqua, Worcester, UK) under white light of 500 µmol·m^−2^·s^−1^ at 25 °C. To study the effect of light wavelength, algae were cultivated in Algem photobioreactor under red or blue light of 500 µmol·m^−2^·s^−1^ at 25 °C. To study the effect of herbicides, cultures were treated with either 5 µM of norflurazon or 20 µM of chlorpropham for 48 h before carotenoid extraction. All the experiments were performed in triplicate.

### 4.2. Standards and Solvents

Phytoene standard (mixture of isomers (E/Z), 95% purity) was purchased from Sigma–Aldrich (Merck KGaA, Darmstadt, Germany). Methanol (MeOH) and Methyl tert Butyl Ether (MtBE), both HPLC grade, were purchased from Fischer Scientific UK Ltd. (Loughborough, Leicestershire, UK).

### 4.3. Carotenoids Extraction

The carotenoids were extracted from wet alga biomass as follows: the samples were centrifuged at 3000× *g* for 5 min at 5 °C; 10 mL MeOH-MtBE (80:20) were added to the pellets and sonicated and vortexed for 20 s; extracts were clarified at the centrifuge then filtered (0.20 µm filter) into amber HPLC vials before HPLC analysis.

*15-cis* phytoene was also extracted from ripe tomatoes (purchased from Waitrose and Partners, UK) to be used as control material because it favourably accumulates the compound at high concentration.

### 4.4. HPLC Analysis

High-Performance Liquid Chromatography equipped with Diode-Array Detection (HPLC-DAD; Agilent Technologies 1200 series, Agilent, Santa Clara, CA, United States), on-line degasser, a quaternary pump system, a YMC30 250 × 4.9 mm I.D S-5µ column (YMC, Europe GmbH) was used to resolve the phytoene and the phytofluene from the carotenoid extracts. The column temperature was set at 15 °C, the gradient solvent system was MeOH (A): MtBE (B) running at 80% A for the first 10 min at a flow rate of 1 mL/min, then increased to 100% A for the next 10 min at a flow rate of 0.5 mL/min before going back to the initial conditions at 20 min. Total run time was 45 min. The absorbance at four wavelengths (276, 282, 293 and 350) was monitored. All data were acquired and analysed with Chemstation for LC System software. Phytoene extracted from ripe tomatoes was stereo-mutated similarly to Melendez-Martinez et al. [29]. The sample extracted was heated at 75 °C for a total of 240 min and it was then injected in the HPLC system for analysis. Methods parameters were as follow: the column was set at 15 °C, the isocratic solvent system was MeOH (A): MtBE (B) running at 88% A at a flow rate of 0.5 mL/min.

### 4.5. UPC^2^ Analysis

Supercritical, or near-critical, CO_2_ was used as the primary eluent in the mobile phase with a MeOH cosolvent gradient to resolve the phytoene from the total-tomato and *D. salina* extracts using a seven-minute gradient. Supercritical fluid chromatography is very powerful when it comes to identifying compounds based on their retention time, as it allows shorter run times than HPLC and therefore a decrease in shifts in the retention time among runs. The UPC^2^ (Waters Limited, Herts, UK) was equipped with a binary solvent delivery pump, auto-sampler, column oven, photodiode array detector and back-pressure regulator. Mobile phase A was CO_2_ and mobile phase B was MeOH with 0.1% formic acid (*v*/*v*). The gradient was held at 5% B for 0.5 min, then increased to 50% B at 4.9 min, held at 50% for 0.1 min, ramped to 5% B in 0.1 min, and equilibrated for 1.9 min to give a total run time of 7 min. The binary solvent flow rate was 1.0 mL/min, the isocratic make-up solvent flow was 0.3 mL/min, and the UPC^2^ was operated with a back-pressure of 2000 psi. The column oven was maintained at a temperature of 50 °C and an Acquity UPC^2^ HSS C18 SB (1.8 µm, 3.0 mm × 100 mm) column was used. A sample injection volume of 3.0 µL was used with all samples. The UPC^2^ was fitted with an Acquity PDA detector (190–800 nm) and operated in the wavelength range 210–600 nm with a 1.2 nm resolution. In addition, the absorbance at four wavelengths (276 nm, 282 nm, 293 nm and 450 nm) was monitored with a resolution of 1.2 nm.

### 4.6. Mass Spectrometry Analysis

All mass spectrometry experiments were collected on a Waters Synapt G2 Q-ToF mass spectrometer (Manchester, UK). The instrument was operated in positive ion mode electrospray with a capillary voltage of 2.0 kV and sampling cone voltage was 30 V. Nitrogen at a flow rate of 650 L/h was used as the desolvation gas with a constant desolvation temperature of 350 °C, a cone gas flow rate of 50 L/h, and a source temperature of 130 °C. Data were acquired over the *m/z* range 50–800. An integral LockSpray unit infusing Leucine-Enkephalin peptide into the electrospray sample stream was used to collect reference scans. Scans were performed every 10 s with the reference calibrant introduced at a flow rate of 10 µL/min using the fluidics system of the instrument. Single point lock-mass correction was used for the protonated pseudo-molecular ion at *m/z* 556.2771 (+ve). All data were acquired and analysed with Waters MassLynx v4.1 software (Manchester, UK).

### 4.7. NMR Sample Preparation

Fresh tomato samples and harvested *D. salina* cultures were freeze-dried and a total of 20 g of dried tomato powder and dried algae samples was used for the carotenoid extraction. The procedure described above was modified as follow: MtBE was chosen as solvent for extraction due to its lower polarity index (2.4), hence its higher selectivity for only carotenoids (and lipids) in comparison to the MeOH and ethanol solvents; moreover, its high volatility value allowed shorter evaporation times and minimisation of sample losses. The extracts were injected in the HPLC system to confirm the presence of phytoene. The MtBE extracts were evaporated for one and a half hour, flushed with nitrogen and suspended with deuterated chloroform prior to analysis.

### 4.8. NMR Conditions

The NMR experiments were carried out with a JEOL (Tokyo, Japan) 500 MHz. All the spectra were recorded at ambient temperature. Proton and carbon spectra for carotenoid analysis were referenced to the TMS signal (δ = 0.00 ppm). 1D Proton parameters: ^1^H 45° flip angle of 6.32 µs, spectral width 9.38 kHz, 65K data points, 64 scans and relaxation delay 2s. 1D Carbon parameters: ^13^C 30° flip angle of 3.56 µs, spectral width of 39.31 kHz, 65k data points, 1024 scans and relaxation delay 2s. 2D ^1^H-^13^C pulse field gradient heteronuclear single quantum correlation spectroscopy (HSQC): pulse of 12.64 µs for ^1^H, pulse of 10.7 µs for ^13^C, decoupling, J_constant of 135 Hz, spectral width of 6.26 and 31.45 kHz for the proton and carbon dimensions, respectively; relaxation delay 2s, 2048 total scan and 1024 and 256 data point in f_2_ and f_1_, respectively. Data were acquired and analysed with MNova software (12.0.3, Mestrelab Research S.L., Santiago de Compostela, Spain).

## 5. Conclusions

The present work used different analytical tools to identify and characterize the isomeric composition of phytoene, an increasingly sought after colourless carotenoid and the precursor of all other carotenoids in the carotenoid biosynthetic pathway, in order to evaluate its production from *D. salina* and help understand the synthesis of high content of *9-cis* β-carotene in *D. salina*. Results show *15-cis* phytoene is the major isomer in *D. salina* (>98% of total phytoene) while *all-trans* phytoene presents in trace amount. *9-cis* phytoene has not been detected in any of the *D. salina* extracts.

## 6. Patents

WO2019097219-Production of *Dunaliella* (2019). Harvey PJ, Xu Y.; WO/2018/141978 Algal strains (2018). Schroeder D, Harvey PJ, Xu Y.

## Figures and Tables

**Figure 1 molecules-25-01880-f001:**
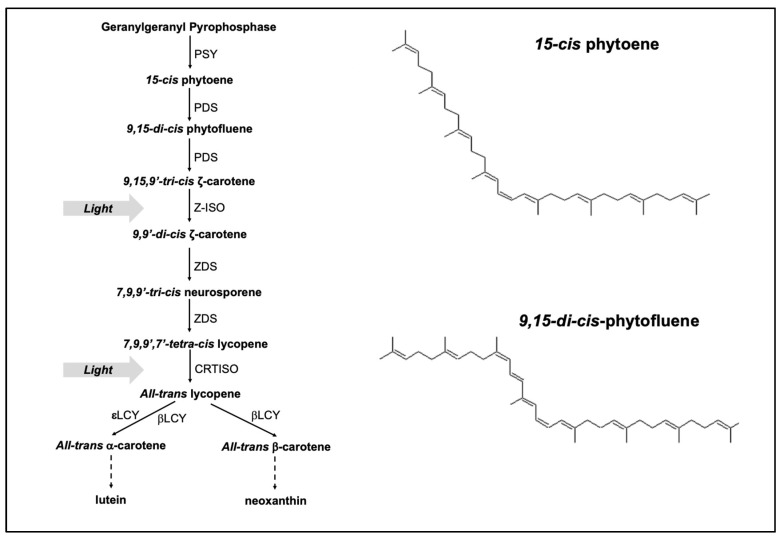
A generally acknowledged biosynthetic pathway of carotenoids in plants and algae [8,9,10]. PSY, phytoene synthase; PDS, phytoene desaturase; Z-ISO, ζ-carotene isomerase; ZDS, ζ-carotene desaturase; CRTISO, carotenoid isomerase; εLCY, lycopene ε-cyclase; βLCY, lycopene β-cyclase; Chemical structures of *15-cis* phytoene and *9,15-di-cis*-phytofluene (ACD/ChemSketch Software) are also shown in the figure.

**Figure 2 molecules-25-01880-f002:**
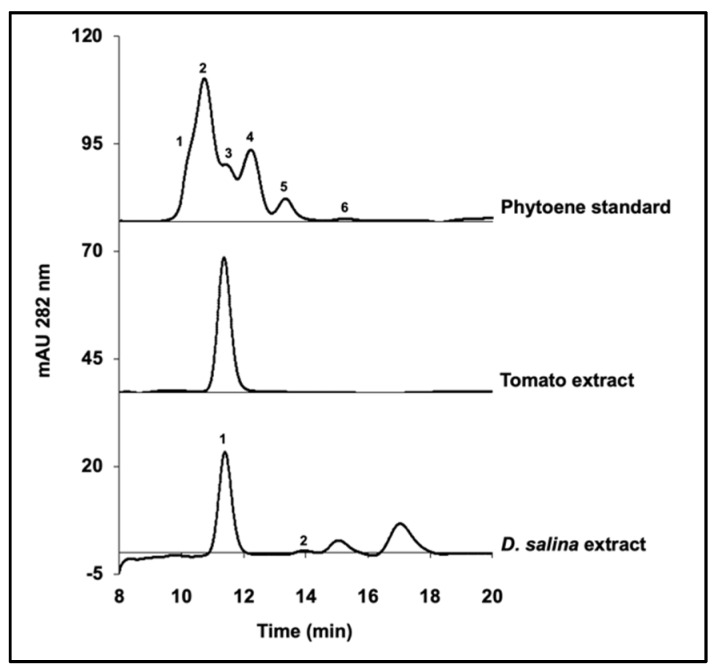
HPLC analysis of phytoene standard purchased from Sigma, phytoene extracts from ripe tomatoes and phytoene extracts from *D. salina* cultures.

**Figure 3 molecules-25-01880-f003:**
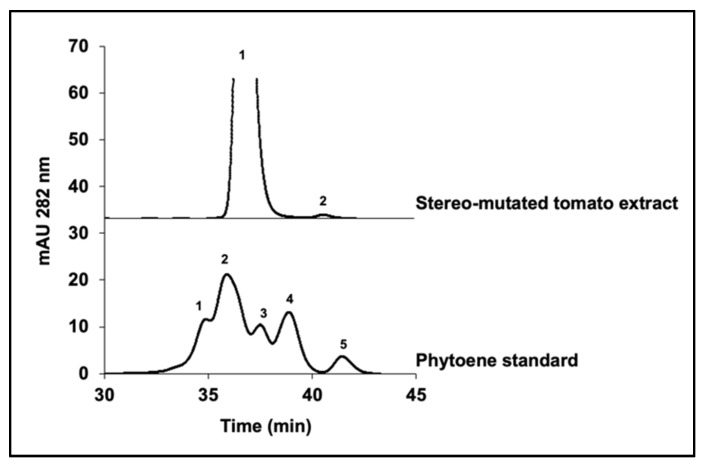
HPLC chromatograms of stereo-mutated phytoene extracts from ripe tomatoes compared to phytoene standard mixture, only showing the region of interest.

**Figure 4 molecules-25-01880-f004:**
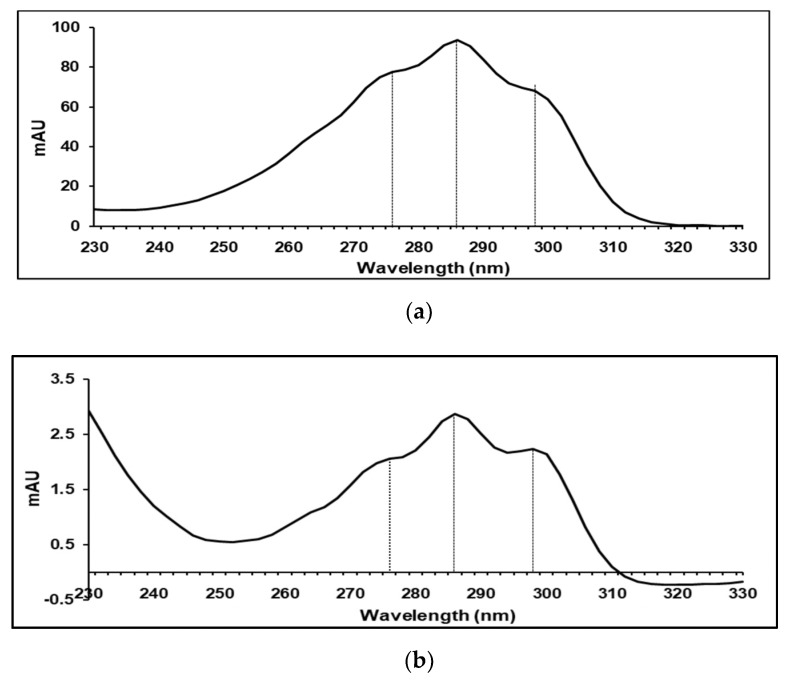
Absorption spectra of (**a**) *15-cis* and (**b**) *all-trans* phytoene analysed by HPLC-diode array in MeOH/MTBE (8/2) solvent with maximum absorption at 275, 285 and 297 nm.

**Figure 5 molecules-25-01880-f005:**
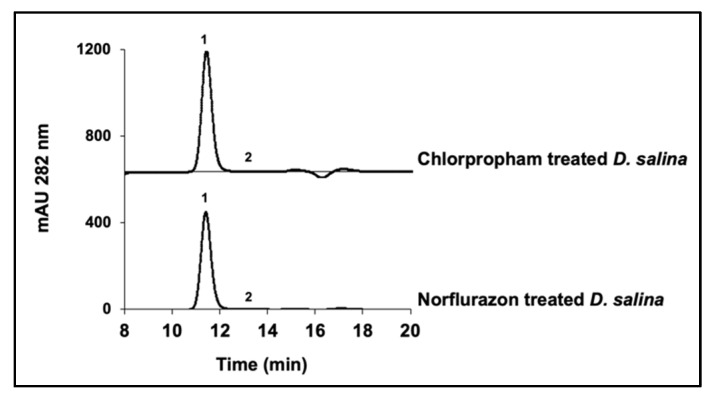
HPLC chromatograms of phytoene extracts from *D. salina* cultures treated with 20 µM of chlorpropham or 5 µM of norflurazon, only showing the region of interest in the chromatogram.

**Figure 6 molecules-25-01880-f006:**
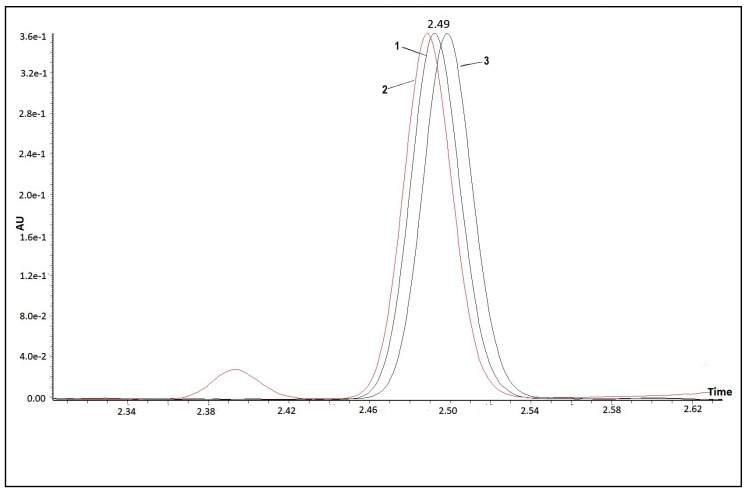
UPC^2^ chromatogram, showing phytoene extracted from ripe tomatoes (peak 1), Norflurazon (peak 2) and chlorpropham (peak 3) treated *D. salina* cultures.

**Figure 7 molecules-25-01880-f007:**
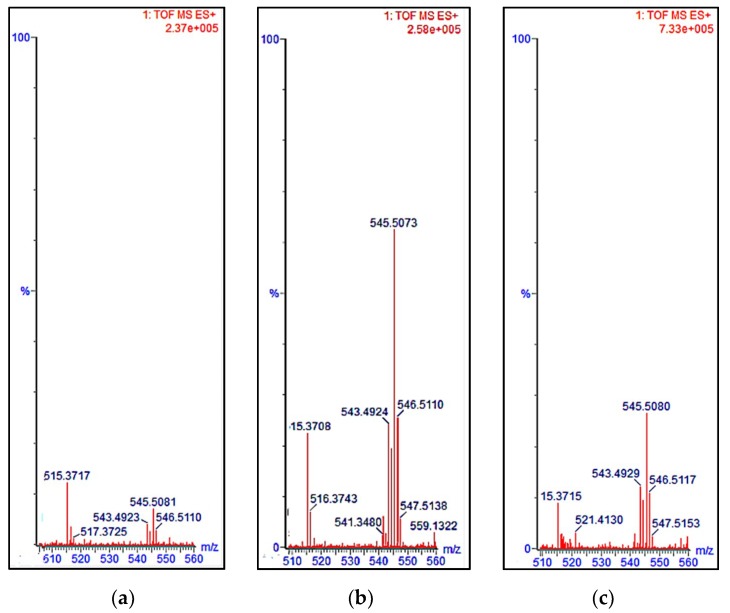
Region of mass spectra showing the molecular mass of phytoene derived from (**a**) ripe tomatoes (545.5081), (**b**) chlorpropham-treated *D. salina* (545.5073) and (**c**) norflurazon-treated *D. salina* (545.5080)*,* in positive ionisation mode [M + H]^+^.

**Figure 8 molecules-25-01880-f008:**
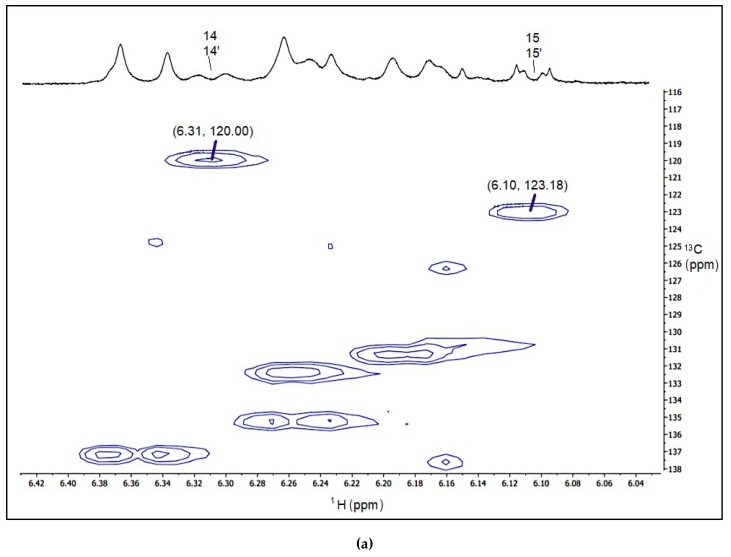
^1^H–^13^C-NMR spectra displayed as heteronuclear single quantum correlation spectra (HSQC) to show the chemical shifts values (ppm) for the carbon atoms C14, C15, C15′ and C14′ of phytoene extracted from (**a**) tomato, (**b**) chlorpropham and (**c**) norflurazon treated *D. salina* extracts. Numbers above the ^1^H-NMR spectra indicate the carbon number of phytoene associated with the resonance.

**Figure 9 molecules-25-01880-f009:**
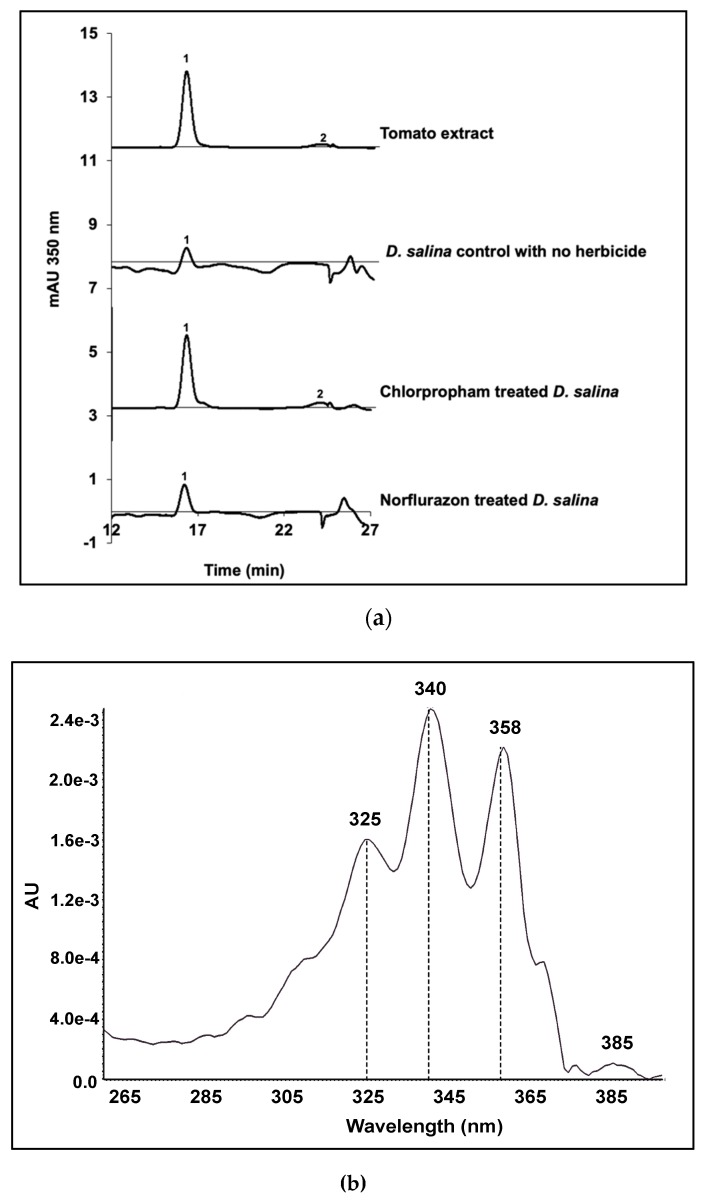
(**a**) HPLC analysis of phytofluene isomers extracted from ripe tomatoes, control *D. salina* cultures with no herbicide, *D. salina* cultures treated with 20 µM chlorpropham, and *D. salina* cultures treated with 5 µM norflurazon, only showing the region of interest in the chromatogram and (**b**) absorption spectrum of phytofluene in Methanol/MTBE (8/2) obtained from UPC^2^.

**Figure 10 molecules-25-01880-f010:**
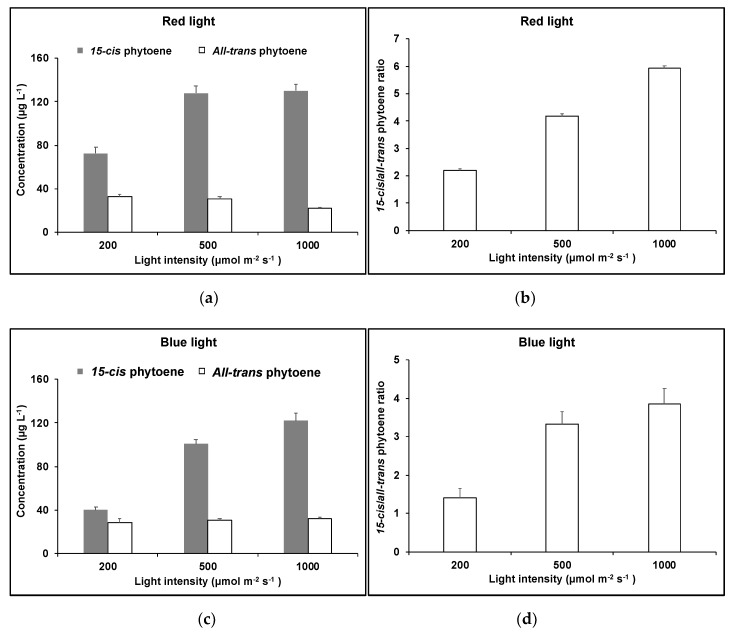
Phytoene isomeric composition in *D. salina* cultivated under (**a**), (**b**) red or (**c**), (**d**) blue light at different light intensities (200, 500 or 1000 µmol·m^−2^·s^−1^) for 48 h.

**Figure 11 molecules-25-01880-f011:**
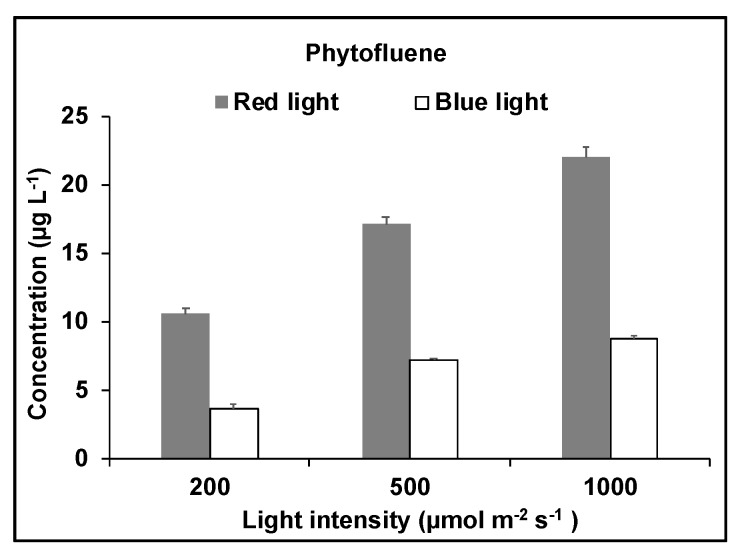
Phytofluene concentration in *D. salina* under red light or blue light at different light intensities (200, 500 or 1000 µmol·m^−2^·s^−1^) for 48 h.

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
