# Peer review of "Stereoisomers of Colourless Carotenoids from the Marine Microalga Dunaliella salina"

_molecules, 2020, doi:10.3390/molecules25081880_

Round 1

Reviewer 1 Report

The authors outline the identification and characterization of the isomeric composition of phytoene from D. salina. The manuscript is well written and the objectives are well defined. The quality of experiments and results are the highlights of the manuscript. There are minor comments. Too many figures are given in the manuscript to explain the same thing. The figures can be combined in one figure to compare the peaks. For example,

Figure 1a, b, and c can be combined in one figure to compare. This will allow a better comparison of the compounds in various sources.  

Similarly Fig. 2, 3, 5, and 9 can be modified to include all figures in one respective figures.

Author Response

We thank the reviewer for these helpful comments.

Figure 1a, b, and c are now combined in one figure in the revised manuscript.

Figures associated with each of Fig. 2, 3, 5, and 9 are now combined into the corresponding figures in the revised manuscript.

Reviewer 2 Report

The authors present a study of stereoisomers of colourless carotenoids from the marine microalga Dunaliella salina.

HPLC, UPC2-MS and NMR were used to characterize and quantify the phytoene isomeric configurations in D. salina.

The manuscript is well written. The manuscript's title, abstract, scheme, tables and figures are adequate to the content. 

The experimental part gives enough details about the synthetic procedures and molecular docking procedures.

However the structure elucidation is based on 1H NMR data, MS(ESI) and elemental anlyses only. The assignment of proton signals is missing. 13C NMR spectra would provide more information about the skeleton structure.

Additional remarks:

  1. 8b and Fig. 8c – please do phase correction in F1 direction more carefully

Author Response

We thank the reviewer for these helpful comments.

We should like to note that the structure elucidation is also based on 13C NMR data and not only 1H NMR data. This is evident in the text, line 179 onwards. Furthermore Figure 8 represents a HSQC 2-D experiment based on the 1H—13C system. We have relabelled the axes to show this more clearly and rewritten the figure legend for Figure 8.

We have now indicated the assignment of proton signals on Figure 8

The F1 direction in Figure 8 is now labelled as 13C ppm to avoid confusion.